# Learning curves theory for hierarchically compositional data with power-law distributed features

**Francesco Cagnetta** [1]   **Hyunmo Kang** [2]   **Matthieu Wyart** [3]

## Abstract

Recent theories suggest that *Neural Scaling Laws* arise whenever the task is linearly decomposed into power-law distributed units. Alternatively, scaling laws also emerge when data exhibit a hierarchically compositional structure, as is thought to occur in language and images. To unify these views, we consider classification and next-token prediction tasks based on probabilistic context-free grammars—probabilistic models that generate data via a hierarchy of production rules. For classification, we show that having power-law distributed production rules results in a power-law learning curve with an exponent depending on the rules' distribution and a large multiplicative constant that depends on the hierarchical structure. By contrast, for next-token prediction, the distribution of production rules controls the local details of the learning curve, but not the exponent describing the large-scale behaviour.

## 1. Introduction

The improvement in the performance of many machine-learning models with the amount of resources, including the number of model parameters and training examples, has been shown to follow a simple power-law behaviour across several orders of magnitude (Hestness et al., 2017; Kaplan et al., 2020). These power laws, known as *neural scaling laws*, are used in practice as a guideline for scaling up resources (Hoffmann et al., 2022; OpenAI, 2023).

Among scaling laws, the *learning curve* describes the im-

---

[1]Theoretical and Scientific Data Science, SISSA, Trieste, Italy [2]Department of Physics, Seoul National University, Seoul, Republic of Korea [3]Department of Physics & Astronomy, Johns Hopkins University, Baltimore, United States. On leave from EPFL, Switzerland. Correspondence to: Francesco Cagnetta <francesco.cagnetta@sissa.it>, Hyunmo Kang <gusah1104@snu.ac.kr>, Matthieu Wyart <mwyart1@jh.edu>.

*Proceedings of the $42^{nd}$ International Conference on Machine Learning*, Vancouver, Canada. PMLR 267, 2025. Copyright 2025 by the author(s).

provement of test performance in the data-limited regime, where model capacity and compute are unlimited, and performance is constrained primarily by the number of training data. A simple approach, based on data memorisation, leads to power-law learning curves under a Zipf, i.e. power-law, distribution of the input data (Hutter, 2021; Michaud et al., 2023). However, this view cannot explain how generalisation performance improves. Alternatively, power-law learning curves appear in kernel regression when the spectrum of the target function in the kernel eigenbasis is itself a power law (Caponnetto & De Vito, 2007). Several theoretical studies of neural scaling laws are indeed based on this result (Spigler et al., 2020; Bordelon et al., 2020; Bahri et al., 2021; Favero et al., 2021; Maloney et al., 2022; Cagnetta et al., 2023; Bordelon et al., 2024; Lin et al., 2024). However, these approaches are restricted to kernel-based approximations of deep learning methods, whose limited power cannot explain the successes of modern language and vision models.

In this respect, recent studies have identified hierarchical generative models such as Probabilistic Context-Free Grammars (PCFGs) as model datasets to explain the difference in performance between deep and kernels/shallow learning methods methods, while still allowing for some analytical understanding (Malach & Shalev-Shwartz, 2018; 2020; Cagnetta et al., 2024; Sclocchi et al., 2025; Mei, 2024; Tomasini & Wyart, 2024; Cagnetta & Wyart, 2024; Garnier-Brun et al., 2024; Sclocchi et al., 2024; Oko et al., 2025). In this work, we combine one such data model—the Random Hierarchy Model (RHM) of (Cagnetta et al., 2024)—with the hypothesis that the features of real datasets (e.g. the words in a text corpus) are Zipf distributed.

### 1.1. Our contributions

- We introduce (section 2) a family of synthetic datasets based on the RHM (Cagnetta et al., 2024), where data are generated from their class labels according to a hierarchy of production rules, mapping high-level features to tuples of lower-level features. While in (Cagnetta et al., 2024) all production rules have the same probability, we consider probabilities obeying Zipf's law, $f_k \propto k^{-(1+a)}$. These datasets model *both* the hierar-

chical and compositional structure *and* the Zipfian distribution of the low-level features of realistic datasets;

- We show (section 4) that the Zipf distribution of features changes the learning curve of classification tasks from a sigmoidal to a power-law shape. In particular, after a large pre-asymptotic phase of size controlled by the hierarchical structure, the classification error of a network trained on $P$ data decays asymptotically as $P^{-a/(1+a)}$;

- For next-token prediction tasks (section 5), where the learning curves of the uniform RHM are already power laws (Cagnetta & Wyart, 2024), the distribution of the features does not change the asymptotic decay, while it controls the curves' local details. This result puts forward the hierarchical structure of language—*not* the Zipf distribution of features—as a prime candidate for explaining the scaling laws of Large Language Models.

### 1.2. Additional related works

There is a growing number of studies using generative models from theoretical linguistics to understand the capabilities of large language models, including $n$-grams (Svete & Cotterell, 2024; Nguyen, 2024; Svete et al., 2024), and regular (Borenstein et al., 2024; Shai et al., 2024) and context-free grammars (Allen-Zhu & Li, 2023; Zhao et al., 2023). All these works concern either expressivity or the interpretability of the representations of trained transformers. (Zhao et al., 2023), in particular, showed that the operations performed by BERT-like transformers resemble well-known algorithms for grammatical inference, and proved that, for PCFG data, these algorithms are optimal solutions of the masked language modelling objective. However, when the training data can be equally explained by a PCFG or a non-hierarchical generative model, neither recurrent language models (McCoy et al., 2020) nor transformer (Ahuja et al., 2024) consistently prefer the hierarchical interpretation. In addition, none of these works study the learning process and the sample complexity.

## 2. Notation and setup

**Hierarchical generative model.** We consider synthetic datasets generated via a probabilistic context-free grammar (PCFG) (Rozenberg & Salomaa, 1997): a collection of symbols and rules that prescribe how to generate input data starting from their label. PCFGs consist of a vocabulary of hidden (*nonterminal*) symbols, a vocabulary of observable (*terminal*) symbols and *production rules* that quantify the probability that one hidden symbol generates tuples of either hidden or observable symbols. Generic PCFGs provide a natural formalism for describing hierarchical structures found in the syntax of natural languages (Pullum & Gazdar,

1982; Joshi, 1985), and have also been used to model semantics (Knuth, 1968) and natural images (Zhu & Mumford, 2006). Here, for the sake of analytical tractability, we make the following simplifying assumptions:

- *i)* the nonterminal symbols are split into $L$ finite vocabularies $(\mathcal{V}_\ell)_{\ell=1,\dots,L}$ of finite size $v$ and $\mathcal{V} \equiv \mathcal{V}_0$ denotes the vocabulary of terminal symbols, or *tokens*;

- *ii)* All the production rules transform one level-$(\ell+1)$ symbol into $s$ level-$\ell$ symbols,

$$\mu^{(\ell+1)} \to \mu_1^{(\ell)}, \dots, \mu_s^{(\ell)}; \qquad (1)$$

- *iii)* There are $m$ *unambiguous* production rules per nonterminal symbol, i.e. two distinct nonterminals cannot generate the same $s$-tuple. These rules are randomly chosen and frozen for a given instance of the RHM;

- *iv)* Each rule can be picked with probability $f_k^{(\ell)}$, with $k = 1, \dots, m$ and $\sum_k f_k^{(\ell)} = 1$ for all $\ell$;

The Random Hierarchy Model (RHM) of (Cagnetta et al., 2024) corresponds to setting $f_k^{(\ell)} = 1/m$ for all $k$'s and $\ell$'s. Here, mimicking the power-law distribution of word frequencies (Corral et al., 2015), we set the production rule distribution to be uniform in all but one layer $\ell$, where it follows a Zipf law (Hutter, 2021; Michaud et al., 2023), $f_k^{(\ell)} \propto k^{-(1+a)}$.[1]

Given an RHM instance, input data are generated by picking a class label $y$ (or level-$L$ symbol) uniformly at random, then picking a production rule emanating from that label and replacing the label with the right-hand side of the production rule. Repeatedly applying the production rules $L$ times yields the input sequence $\boldsymbol{x} = (x_1, \dots, x_d)$, with $d = s^L$. The process can be represented graphically with a tree, as illustrated in Fig. 1, known as the *derivation* of the sequence $\boldsymbol{x}$. Each token $x_i$ is represented as a $v$-dimensional one-hot vector $(x_{i,\mu})_{\mu=1,\dots,v}$, with $x_{i,\mu} = 1$ if $x_i$ encodes for the $\mu$-th element of the vocabulary and $0$ otherwise.

**Learning Setup.** We consider both classification and next-token prediction tasks. For classification, deep convolutional networks (CNNs) are trained to approximate the probability of the root (label) conditioned on the input, $\mathbb{P}\{Y = y | X_1 = x_1, \dots, X_d = x_d\}$. For next-token prediction, we train deep transformers to approximate

---

[1] Having Zipf production rules at all levels would change the input features distribution, as their probability will be given by the product of several Zipf-distributed numbers. Nevertheless, we believe that our theoretical framework and the main conclusion of our paper would still hold: the exponent of the scaling law is controlled by the Zipf distribution of production rules for classification and by the hierarchical structure for next-token prediction.

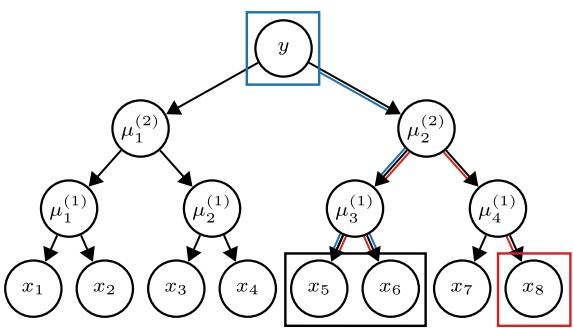

*Figure 1.* Pictorial representation of a *derivation* according to the RHM, with depth $L = 3$ and branching factor $s = 2$. A classification task requires predicting the root label (blue square) from the leaves. The correlations between the 2-tuples of leaves (e.g. $(x_5, x_6)$) and the label $y$ can be used to infer the hidden symbol above the 2-tuple ($\mu_3^{(1)}$ for $(x_5, x_6)$). A next-token prediction task requires predicting the last observable symbol (red square) from the previous $d - 1$. In this case, hidden symbols can be deduced from the correlations of 2-tuples with the last token $x_d$.

the conditional probability of the last token given the other $d - 1$, $\mathbb{P}\{X_d = y | X_1 = x_1, \ldots, X_{d-1} = x_{d-1}\}$. In both cases, training proceeds by updating the model's parameters via gradient descent over the empirical cross-entropy loss. Numerical experiments are performed in PyTorch (Paszke et al., 2019), with the code available at https://github.com/fracagnetta/random-hierarchy-model. The specifics of the machine learning models, including training hyperparameters and computational resources, are designed to reflect the data-limited regime, as detailed in Appendix A.

## 3. Theoretical background

### 3.1. Hutter's theory of learning

In the scenario proposed in (Hutter, 2021), each datum consists of a single discrete feature $k = 1, \ldots, \infty$, which uniquely determines the label. Data are drawn i.i.d. according to some probability distribution $f_k$, and correct classification requires that the feature $k$ has appeared at least once in the training set. The resulting test error is the (average) probability that the test feature has not been seen during training,

$$\varepsilon(P) = \sum_{k=1}^{\infty} f_k (1 - f_k)^P, \qquad (2)$$

where $P$ denotes the number of training examples. Assuming a Zipf distribution of the features, $f_k \propto k^{-(1+a)}$, leads to the asymptotic power-law $\varepsilon(P) \sim P^{-a/(1+a)}$ (Hutter,

2021).

### 3.2. Learning the Random Hierarchy Model

Unlike Hutter's model, RHM data are characterised by the hidden hierarchical structure, i.e. the sequence of production rules used during generation. Multiplying the probabilities of all these production rules yields the conditional probability of the input sequence, given the root label, $\mathbb{P}\{X_1 = x_1, \ldots, X_d = x_d | Y = y\}$. As shown in (Cagnetta et al., 2024), production rules can be inferred from the correlations between the root label $Y$ (blue box in Fig. 1) and $s$-tuples of contiguous input tokens $\boldsymbol{X}_j = (X_{(j-1)s+1}, \ldots, X_{js})$ (black box in Fig. 1),

$$C_j(y, \boldsymbol{\mu}) := \mathbb{P}\{Y = y; X_{(j-1)s+1} = \mu_1, \ldots, X_{js} = \mu_s\} \\ - \mathbb{P}\{Y = y\} \mathbb{P}\{\boldsymbol{X}_j = \boldsymbol{\mu}\}. \qquad (3)$$

Indeed, due to the context-free structure of the generative model, the $C_j(y, \boldsymbol{\mu})$ are identical for all the $s$-tuples $\boldsymbol{\mu}$ generated by the same level-1 nonterminal symbol $\mu^{(1)}$. With sufficient training data, empirical estimates of these correlations can be used to cluster $s$-tuples by their generating nonterminal, enabling a bottom-up reconstruction of the hidden hierarchical structure. This approach was further extended in (Cagnetta & Wyart, 2024) to next-token prediction, where the relevant correlations are those between $s$-tuples of input tokens and the final token of the sequence (red box in Fig. 1), instead of the root label.

The role of correlations in learning the RHM can be formalised via the following

**Assumption 3.1.** *Each production rule is learnt when its effect on correlations can be detected from the training data.*

Then, the learning curve is derived by linking the generalisation performance of a model to the number of production rules that can be inferred from the training data. For instance, in a classification task with uniform production rules, all the rules are learnt once the training set size exceeds $P^* = vm^L$ (Cagnetta et al., 2024). This sample complexity is derived by balancing two sources of variance: the variance between correlations corresponding to different level-1 nonterminals—scaling as $(v^3 m^{L+1})^{-1}$—and the sampling noise induced by finite data, scaling as $(v^2 mP)^{-1}$. When $P \gg P^*$, the correlations become distinguishable, allowing the model to reliably infer level-1 rules. Since the data structure is recursive, resolving the first hidden level of the tree simplifies the problem, allowing reconstruction of the full tree. Consequently, the learning curve exhibits a sigmoidal shape, with an inflection point that scales as $P^* = vm^L$, as confirmed empirically in experiments with deep CNNs.

# 4. Root classification

## 4.1. Nonuniform production rules at the bottom layer

When only level-1 production rules have a nonuniform distribution, it is convenient to factorise the contribution of these rules in the data probability:

$$\mathbb{P}\{X_1 = x_1, \ldots, X_d = x_d | Y = y\} =$$
$$\left(\prod_{j=1}^{s^{L-1}} \mathbb{P}\left\{\boldsymbol{X}_j = \boldsymbol{\mu}_j | X_j^{(1)} = \mu_j^{(1)}\right\}\right) \times$$
$$\mathbb{P}\left\{X_1^{(1)} = \mu_1^{(1)}, \ldots, X_{s^{L-1}}^{(1)} = \mu_{s^{L-1}}^{(1)} | Y = y\right\}. \quad (4)$$

The conditional probability of level-1 nonterminals given the root is equal to the probability of a uniform RHM with $L-1$ layers. The factors inside the brackets, instead, are nothing but the probabilities of level-1 rules: given the *rank* $k(\boldsymbol{\mu}_j)$ of the tuple $\boldsymbol{\mu}_j$, i.e. the index $k \in 1, \ldots, m$ of the unique production rule that generates $\boldsymbol{\mu}_j$, the corresponding probability equals $f_{k(\boldsymbol{\mu}_j)}$. By summing over all the tuples but the $j$-th, we get

$$\mathbb{P}\{\boldsymbol{X}_j = \boldsymbol{\mu} | Y = y\} = f_{k(\boldsymbol{\mu})}\mathbb{P}\left\{\mu_j^{(1)} = \mu_1(\boldsymbol{\mu}) | Y = y\right\}, \quad (5)$$

where $\mu_1(\boldsymbol{\mu})$ denotes the unique level-1 features that generates $\boldsymbol{\mu}$. Summing over $y$ yields a similar result for the probability of $\boldsymbol{\mu}$: $\mathbb{P}\{\boldsymbol{X}_j = \boldsymbol{\mu}\} = f_{k(\boldsymbol{\mu})}\mathbb{P}\left\{\mu_j^{(1)} = \mu_1(\boldsymbol{\mu})\right\}$. Then, by Eq. 3,

$$C_j^{(L)}(\boldsymbol{\mu}, y) = f_{k(\boldsymbol{\mu})} C_j^{(L-1)}(\mu_1(\boldsymbol{\mu}), y), \quad (6)$$

where the dependence on the number of layers has been made explicit.

Since the layers above the first have uniform production rules, we can use the results of (Cagnetta et al., 2024) for the variance of $C_j^{(L-1)}(\mu_1, y)$, i.e. $(v^3 m^{L-1})^{-1}$, yielding a variance of $f_{k(\boldsymbol{\mu})}^2/(v^3 m^{L-1})$ for $C_j^{(L)}(\boldsymbol{\mu}, y)$. In contrast with the uniform case, the variance depends on the rank $k(\boldsymbol{\mu})$ of the low-level tuple $\boldsymbol{\mu}$. However, it correctly reduces to the uniform case result when $f_{k(\boldsymbol{\mu})} = 1/m$. The sampling noise is also affected by the probability of the production rules, as the probability of data with $\boldsymbol{X}_j = \boldsymbol{\mu}$ is proportional to $f_{k(\boldsymbol{\mu})}$ and the variance due to sampling is proportional to the event's probability. This effect is equivalent to replacing $P$ by $P/(f_{k(\boldsymbol{\mu})}m)$, so that the variance due to the sampling noise reads $f_{k(\boldsymbol{\mu})}/(v^2 P)$. Consequently, the sample size necessary to resolve the correlations of the tuple $\boldsymbol{\mu}$ with the label is $P^*(\boldsymbol{\mu}) = v m^{L-1}/(f_{k(\boldsymbol{\mu})})$.

Ranking all the low-level tuples by the probability of the corresponding production rules yields a sequence of $m$ sample complexities $P_k^* = v m^{L-1}/(f_k)$. To estimate the learning curve, we assume that, following 3.1, when $P > P_k^*$ the model can correctly classify data consisting of tuples with

probability higher than $f_k$. In other words, the model classifies the input correctly if and only if all the $s^{L-1}$ input patches are resolvable. [2] The resulting test error reads

$$\varepsilon(P) = 1 - \left(\sum_{k | P_k^* < P} f_k\right)^{s^{L-1}}. \quad (7)$$

When $P \gg P_1^* \simeq v m^{L-1}$, this expression implies, as shown in Appendix B,

$$\varepsilon(P) \simeq s^{L-1} \left(\frac{P}{v m^{L-1}}\right)^{-a/(1+a)}. \quad (8)$$

## 4.2. Nonuniform distribution of production rules at arbitrary layer

When the nonuniform distribution of production rules affects an arbitrary layer $\ell \neq 1$, the probabilities of low-level tuples can be decomposed as sums of conditional probabilities over a specific choice of production rules. As a result, the correlations of Eq. 3 can also be written as a sum of contributions due to production rules of a given probability $f_k^{(\ell)}$. In analogy with Eq. 6, these contributions can be split into a factor of $f_k^{(\ell)}$ times a label-to-level-$\ell$ correlation and a level-$(\ell-1)$-to-input correlation. These two correlations depend solely on uniformly distributed production rules, thus they can be computed again using the results of (Cagnetta et al., 2024), and the arguments of the previous paragraphs apply.

## 4.3. Empirical scaling laws of deep CNNs

We confirm Eqs. (7) and (8) empirically by measuring the learning curves of deep CNNs trained on root classification of the RHM for *i)* varying Zipf exponent $a$ and *ii)* varying the number $m$ of production rules per symbol. The results, shown in Fig. 2 and Fig. 3 left panels, respectively, display remarkable agreement with our predictions. Furthermore, the right panel of Fig. 2 shows that having Zipf-distributed production rules in the first instead of the last RHM layer leads to a similar learning curve. The right panel of Fig. 3, instead, highlights that the asymptotic scaling law $P^{-1/(1+a)}$ kicks in for $P \gg v m^{L-1}$—the sample complexity of a uniform RHM with $L-1$ layers. We conclude that, for classification, the hierarchical structure controls the size of the preasymptotic phase, whereas the Zipf distribution determines the asymptotic decay. Further empirical tests are presented in Appendix C.

---

[2]We remark that, depending on the $m/v^{s-1}$, the root might still be inferred without resolving all the patches. The corresponding probability was derived in (Sclocchi et al., 2024) for an optimal decoder. However, if $m = v^{s-1}$ as in Fig. 2 and 3, changing one patch changes the root label with high probability, implying that all patches need to be resolved.

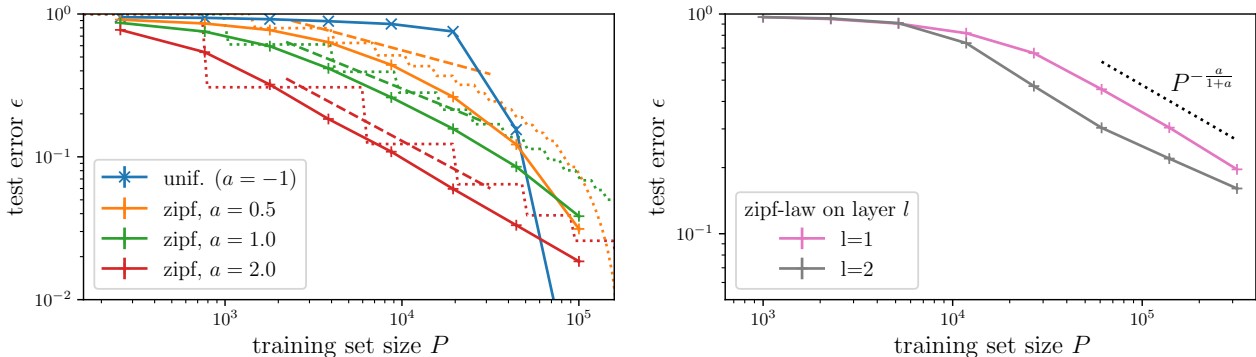

*Figure 2.* **Left:** Learning curves of 3-layers CNNs trained on RHM data with $L=2$, $s=2$, $v=m=25$ and Zipf exponent $a$ indicated in caption. Solid lines are the empirical learning curves whereas dotted lines are predictions from Eq. (7). The dashed line represents the scaling law $\epsilon \sim P^{-a/(1+a)}$. **Right:** As in the left panel, but $v=m=100$. Here $a$ is fixed and the layer where production rules are Zipf-distributed changes. The black dotted line represents the scaling law $\epsilon \sim P^{-a/(1+a)}$.

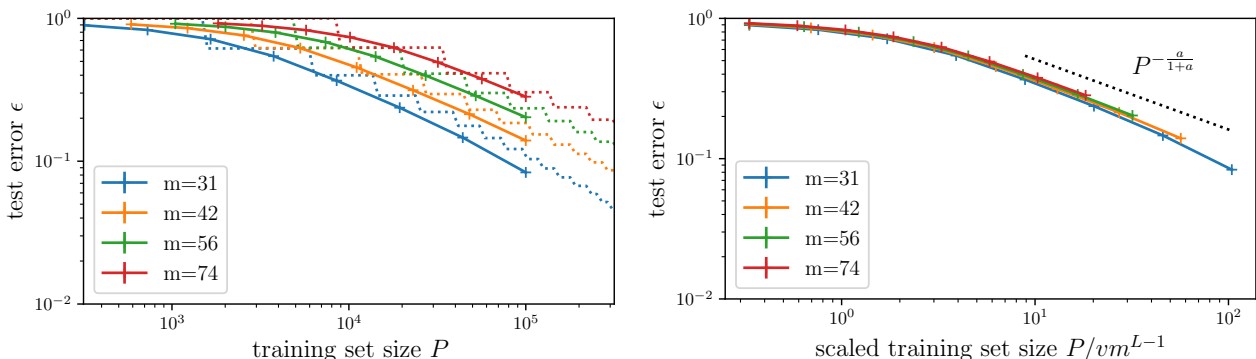

*Figure 3.* **Left:** Learning curves in the same setting as Fig. 2, with Zipf exponent $a=1$ and $m=v$ indicated in caption. Solid lines are the empirical learning curves whereas dotted lines are predictions from Eq. (7). **Right:** all the curves collapse when rescaling the x-axis by $vm^{L-1}$—the sample complexity of an RHM with uniform production rules and $L-1$ layers. The black dotted line represents the scaling law $\epsilon \sim P^{-a/(1+a)}$.

## 5. Next-token prediction

In this section, we use Assumption Theorem 3.1 to derive the learning curves of next-token prediction tasks. Remarkably, the effect on the production rules distribution differs from the classification case, as the exponent governing the large-scale behaviour of the curves depends entirely on the hierarchical structure and not on Zipf's law exponent $a$.

**Uniform RHM.** In contrast with classification, next-token prediction tasks are learnt in a stepwise fashion (Cagnetta & Wyart, 2024). Each step corresponds to learning all the rules associated with the tree up to one of the ancestors of the last token $X_d$ (e.g. $\mu_4^{(1)}$, $\mu_2^{(2)}$ and $y$ for the tree in Fig. 1). At the $\ell$-th step, the model learns the depth-$\ell$ tree generated by the level-$\ell$ ancestor of $X_d$. As in classification, the hidden variables forming these subtrees can be deduced from correlations. As the root label is not provided, we consider correlations between $X_d$ and the $j$-th $s$-tuple of input tokens,

$$C_j(\boldsymbol{\mu}, \nu) := \mathbb{P}\{\boldsymbol{X}_j = \boldsymbol{\mu}, X_d = \nu\} - \mathbb{P}\{\boldsymbol{X}_j = \boldsymbol{\mu}\}\,\mathbb{P}\{X_d = \nu\}. \quad (9)$$

In the uniform RHM, these correlations are random variables over different dataset realisations, having 0 mean and variance (Cagnetta & Wyart, 2024),

$$\langle (C_j(\boldsymbol{\mu}, \nu)^2) \rangle = \frac{1}{v^2 m} \frac{(1 - m/v^{s-1})}{vm^{2\ell-1}}, \quad (10)$$

where $\ell$ is the level of the lowest common ancestor of $X_d$ and $\boldsymbol{X}_j$. Comparing this variance with that due to sampling—given by $(v^2 m P)^{-1/2}$—yields a sequence of sample complexities $P_\ell$ to learn the production rules within the subtree descending from the level-$\ell$ ancestor of $X_d$. When $P \gg P_\ell$, the model outputs the $s^\ell$-gram approxima-

tion of the last-token probability,

$$\mathbb{P}\left\{X_d = x_d | X_{d-(s^\ell-1)} = x_{d-(s^\ell-1)}, \ldots, X_{d-1} = x_{d-1}\right\}. \tag{11}$$

Combining the scaling of $P_\ell$ with $\ell$ with that of the average cross-entropy losses of the $s^\ell$-grams, $\mathcal{L}_\ell$, yields the *scaling law* (Cagnetta & Wyart, 2024)

$$\mathcal{L}(P) \sim P^{-\log(m/v^{s-1})/(2\log m)}, \tag{12}$$

where $\sim$ implies that $P$-independent factors are neglected.

### 5.1. Nonuniform production rules at the bottom layer

**First step via memorisation.** In the first step the model learns that the value of $X_d$ is affected by all the other tokens in the last $s$-tuple, $(X_{d-(s-1)}, \ldots, X_{d-1})$, and outputs the $s$-gram next-token probability $p(x_d|x_{d-(s-1)}, \ldots, x_{d-1})$. As there are no hidden variables that summarise the effect of the context $(X_{d-(s-1)}, \ldots, X_{d-1})$ on $X_d$, the simplest strategy is to memorise all the possible $s$-tuples. Hence, the first step can be described within the framework of (Hutter, 2021). There are $mv$ $s$-tuples, split into $m$ groups according to the probability $f_k$ of the corresponding production rule. Assuming that the level-1 nonterminal is uniformly distributed over the $v$ vocabulary entries, which is true in the limit of large $m$ (Cagnetta et al., 2024), then each $s$-tuple occurs with probability $f_k/v$ in the data. Therefore, the model learns the $v$ most frequent rules with $v/f_1$ data, then the second most frequent with $v/f_2$ data, until converging to the $s$-gram probability with $v/f_m \simeq vm^{1+a}$ data. This scenario is confirmed in Fig. 4, showing the empirical learning curves of a one-layer transformer trained to predict the last token of a depth-1 RHM. In the left panel, a green vertical dashed line marks the sample complexity $v$ required to learn the most frequent rules. Notice that, when $a \geq 1$, $v/f_1 \simeq v$, indeed at $P \simeq v$, the test loss departs from the initial value $\log v$, corresponding to a random guess of the last token, and slowly converges towards the average entropy of the $s$-gram probability $\mathcal{L}_1(a)$, which can be computed exactly knowing the rules of the RHM. As in (Hutter, 2021), the convergence to the average $s$-gram entropy follows the power law $\sim P^{-a/(1+a)}$ (right panel).

**Further steps via reconstruction of the tree.** Starting from the second step, the simple memorisation strategy can be improved by replacing all the $s$-tuples of visible tokens with the corresponding hidden variables. Let us consider, for instance, the data structure depicted in Fig. 1. The pair $(x_5, x_6)$ appears in the $s^2$-gram $p(x_8|x_5, x_6, x_7)$. Since there is only one level-1 nonterminal $\mu^{(1)}$ generating $(x_5, x_6)$, $p(x_8|x_5, x_6, x_7) = \tilde{p}(x_8|\mu^{(1)}(x_5, x_6), x_7)$. Therefore, a model having access to the level-1 nonterminals does not need to memorise all combinations of $(x_5, x_6)$ and

could reach the $s^2$-gram approximation with fewer training examples. To derive the sample complexity we resort again to Assumption 3.1. As we found in section 4, due to the difference in production rule probabilities, the statistics of the tuple-token correlations $C_j(\boldsymbol{\mu}, \nu)$ depend not only on the level of the lowest common ancestor $\ell$, but also on the rank $k(\boldsymbol{\mu})$ of the tuple $\boldsymbol{\mu}$. In particular, the variance of $C_j(\boldsymbol{\mu}, \nu)$ over RHM realisations reads,

$$\left\langle (C_j(\boldsymbol{\mu}, \nu))^2 \right\rangle = \left(f_{k(\boldsymbol{\mu})}\right)^2 \left(\sum_{k'=1}^m f_{k'}^2\right) \frac{(1 - m/v^{s-1})}{v^3 m^{2\ell-3}}, \tag{13}$$

where $\ell \geq 2$. The detailed derivation of Eq. 13 is provided in Appendix D. The term $\sum_{k'} f_{k'}^2$ is the inverse participation ratio of the production rules distribution: it ranges from $1/m$ to 1. It measures the localisation of the distribution $f_k$, with $1/m$ corresponding to a uniform distribution and 1 to the case where all the probability is concentrated on 1 production rule. Therefore, Eq. 10 is recovered in the uniform case. For a general Zipf exponent $a > 0$, $\sum_{k'} f_{k'}^2$ converges to some finite, $a$-dependent number. The variance due to sampling is also affected by the production rules distribution, going from $1/(mv^2 P)$ of the uniform case to $f_k(\boldsymbol{\mu})/(v^2 P)$. Equating the variance due to sampling to Eq. 13 gives the following sample complexities, for reconstructing the $k$-th rule of the $\ell$-th level with $\ell \geq 2$:

$$P_{\ell,k} = \frac{vm^{2\ell-3}}{(1 - m/v^{s-1})f_k \left(\sum_{k'} f_{k'}^2\right)}. \tag{14}$$

Notice that, despite the acquired dependence on the rule rank $k$, the dependence on the level $\ell$ is the same as in the uniform case, $P_\ell \sim m^{2\ell}$.

**Average cross-entropy of the $s^\ell$-grams.** According to Assumption 3.1, we expect that, when $P > P_{\ell,m}$, a deep machine-learning model trained on $P$ performs at least as well as the $s^\ell$-gram from Eq. 11. Conversely, if $P < P_{\ell,1}$, the model can at most match the performance of the $s^{\ell-1}$-gram. Therefore, we can combine the average $s^\ell$-grams cross-entropies with the $P_{\ell,k}$'s to determine the scaling law of next-token prediction. These cross-entropies are given by

$$\mathcal{L}_\ell = \left\langle \mathbb{E}_{\boldsymbol{x}} \left[ -\log p(x_d|x_{d-(s^\ell-1)}, \ldots, x_{d-1}) \right] \right\rangle. \tag{15}$$

Let us start by rewriting Eq. 15 using the tree structure of the generative model. The resolved context window $(x_{d-(s^\ell-1)}, \ldots, x_{d-1})$ consist of $s^{\ell-1} - 1$ complete $s$-tuples and one incomplete tuple For instance, in Fig. 1 with $\ell = 2$, the context window $(x_5, x_6, x_7)$ contains the 2-tuple $(x_5, x_6)$ and $x_7$. Due to the unambiguity of the rules, each complete $s$-tuple corresponds to a unique level-1 nonterminal ($\mu_3^{(1)}$ in the example of the figure). Since $m < v^{s-1}$, the generative process can only generate a fraction of the

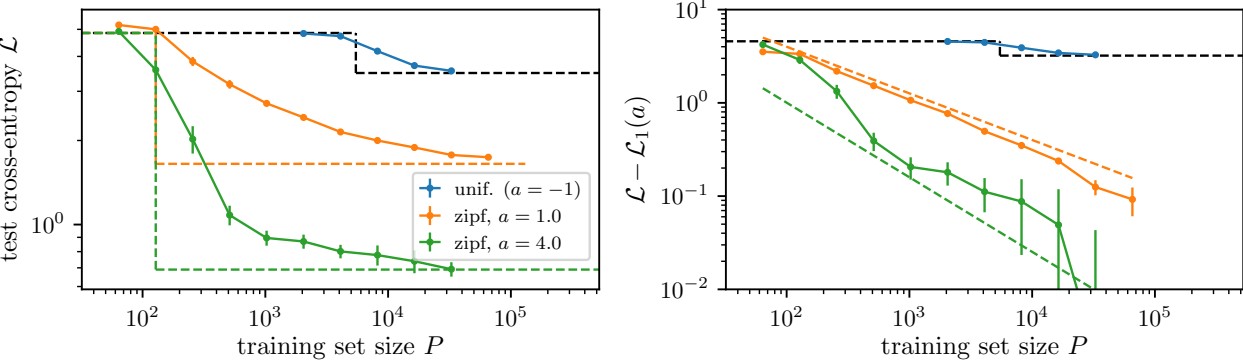

*Figure 4.* **Left:** Empirical learning curve of one-layer transformers trained for next-token prediction on RHM data with $L = 1$, $s = 2$, $v = 128$, $m = 32$ and Zipf exponent $a$ as in the key. Vertical dashed lines mark the sample sizes required to learn the most frequent rules: $vm$ in the uniform case (equivalent to setting $a = -1$ in Zipf's law) and $v$ with Zipf-distributed production rules. The leftmost horizontal dashed lines denote the test loss of the trivial prediction where the next-token probability is uniform over the vocabulary, $\mathcal{L}_0 = \log v$. The rightmost horizontal dashed lines denote the average cross-entropy of the $s$-gram approximation, $\mathcal{L}_1(a)$. **Right:** subtracting $\mathcal{L}_1(a)$ reveals the power-law decay $P^{-a/(1+a)}$, highlighted by coloured dashed lines.

possible sequences of $s^{\ell-1}$ level-1 nonterminal symbols. As a result, knowledge of the $s^{\ell-1} - 1$ nonterminals above the complete $s$-tuples of the context window constrains the parent node of the last token ($\mu_4^{(1)}$ in the figure). Denoting with $\mathcal{V}_c(x_{d-(s^\ell-1)}, \ldots, x_{d-s})$ the set of the level-1 symbols *compatible* with such constraint,

$$
\mathcal{L}_\ell = \left\langle \mathbb{E}_{\boldsymbol{x}} \left[ -\log \frac{1}{\mathcal{Z}(x_{d-(s^\ell-1)}, \ldots, x_{d-s})} \times \right. \right.
$$

$$
\left. \left. \sum_{\mu^{(1)} \in \mathcal{V}_{c,\ell}} p(x_d | x_{d-(s-1)}, \ldots, x_{d-1}; \mu^{(1)}) \right] \right\rangle, \quad (16)
$$

where $\mathcal{Z}(x_{d-(s^\ell-1)}, \ldots, x_{d-s})$ denotes the normalisation $\sum_{\mu^{(1)}, x_d} p(x_d | x_{d-(s-1)}, \ldots, x_{d-1}; \mu^{(1)})$. Notice that, as the $s^{\ell-1} - 1$ tuples in $(x_{d-(s^\ell-1)}, \ldots, x_{d-s})$ can be replaced with the corresponding level-1 nonterminals, $\mathcal{V}_{c,\ell}$ depends only on the highest $L - 1$ levels of the tree. Therefore, it can be thought of as the set of terminal tokens compatible with the context of size $s^{\ell-1} - 1$ in a uniform RHM with $L - 1$ layers The average size of this set over RHM realisations is given by (Cagnetta et al., 2023)

$$
\langle |\mathcal{V}_{c,\ell}| \rangle = \frac{1}{1 - m/v^{s-1}} + v \left( \frac{m}{v^{s-1}} \right)^{\ell-1}. \quad (17)
$$

In the limit where $1 \ll m \ll v^{s-1}$, $\langle |\mathcal{V}_{c,\ell}| \rangle$ approaches 1 as $\ell$ grows. In this limit, and for large $\ell$, we can neglect the cases where $|\mathcal{V}_{c,\ell}| > 2$. As a result, most of the data will give a null contribution to Eq. 16, since the context uniquely determines the content of the last token as $x_d$. The only data yielding a positive term are those where there is one

other vocabulary entry, $\tilde{x} \neq x_d$, compatible with the context. Hence, we estimate $\mathcal{L}_\ell$ as the fraction of data with a positive contribution to the entropy, times this contribution. The fraction of data can be derived by examining the possible sources of ambiguity: either $|\mathcal{V}_{c,\ell}| = 1$ and there are 2 distinct production rules compatible with the context and the level-1 symbol, $|\mathcal{V}_{c,\ell}| = 2$ and the extra production rule can also come from a different level-1 symbol. The entropic contribution per datum is the logarithmic sum of two probabilities taken from the set $(f_k)$ of production rules probabilities, thus it depends on the distribution of low-level production rules but not on the other levels of the hierarchy. The derivation of $\mathcal{L}_\ell$ following this argument is presented in Appendix E and leads to

$$
\mathcal{L}_\ell \xrightarrow[1 \ll m \ll v^{s-1}]{} H_{2,a,m} \left( \frac{m/v^{s-1}}{1 - m/v^{s-1}} + v \left( \frac{m}{v^{s-1}} \right)^\ell \right), \quad (18)
$$

where $H_{2,a,m}$ denotes the average entropic contribution of data with 2 compatible entries of the last token. Notice that Eq. 18 displays the same dependence on $\ell$ as the uniform production rules case (Cagnetta & Wyart, 2024).

We can confirm this argument by computing the values of the $\mathcal{L}_\ell$'s for a given set of rules exactly via Eq. 16. The results of this calculation averaged over RHM realisations with fixed $s$, $v$, $m$ and varying production rules distribution are shown in Fig. 5, and confirm that the approach of $\mathcal{L}_\ell$ to the limiting, residual cross-entropy $\mathcal{L}_\infty(a)$ is independent of $a$ and follows the same scaling as the uniform case, shown as a black dashed line.

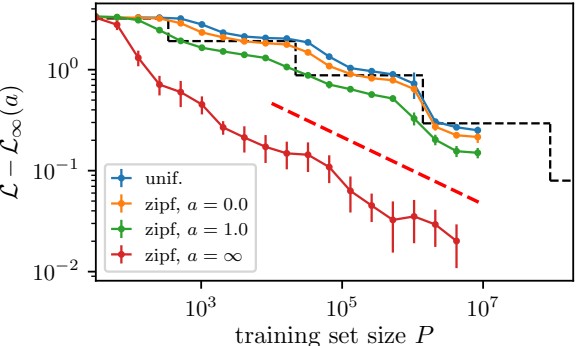

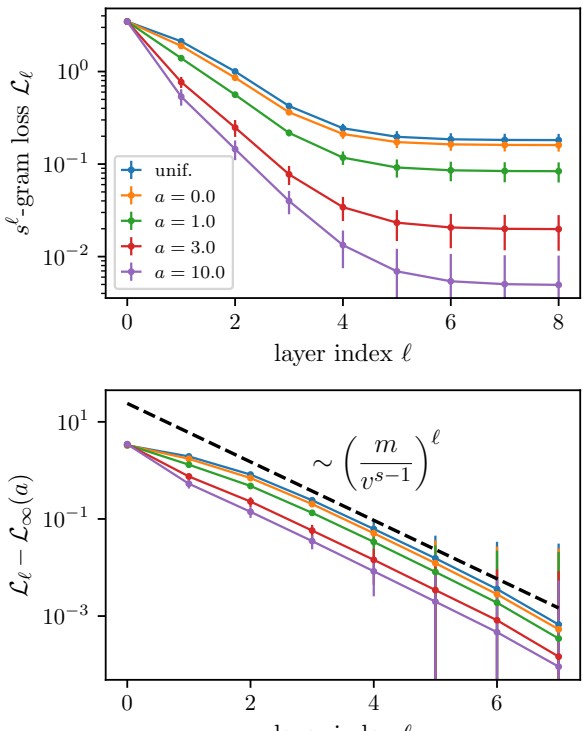

*Figure 5.* Average cross-entropies of the $s^\ell$-grams versus $\ell$, for RHM datasets with $s=2$, $v=32$, $m=8$, with the colour denoting the Zipf exponent. The points are obtained by averaging the cross-entropies over 32 independent realisations of the RHM. The cross-entropies of the uniform production rules case are shown in blue for comparison. For all $a$'s, the cross-entropies $\mathcal{L}_\ell$ decay with $\ell$ towards some $a$-dependent value $\mathcal{L}_\infty(a)$ (**Top** panel). However, the approach to $\mathcal{L}_\infty(a)$ is independent of $a$ (**Bottom** panel) and follows the behaviour of the test loss bound derived in (Cagnetta & Wyart, 2024) in the uniform case (black dashed line).

### 5.2. Empirical scaling laws of deep transformers

Since both the sample complexities and the loss plateaus display the same behaviour with $\ell$ as the uniform case, we predict that the power-law distribution of level-1 production rules does not change the scaling law of the problem in Eq. 12. We confirm this prediction empirically in Fig. 6, showing the learning curves of deep transformers trained on RHM data for several distributions of production rules, from the uniform case of (Cagnetta & Wyart, 2024) to the extreme case where all the probability mass of the $f_k$'s is concentrated on one production rule. The asymptotic decay of the learning curves agrees with Eq. 12, highlighted in the figure by a red dashed line.

*Figure 6.* Empirical scaling laws of depth-4 transformers trained on RHM next-token prediction with $L=4$, $s=2$, $v=32$, $m=8$ and varying $a$. The limit $a \to \infty$ corresponds to having only one production rule per level-1 nonterminal symbol. The red dashed line is a guide to the eye for the asymptotic decay of Eq. 12.

## 6. Conclusions

We studied how the scaling laws of deep networks trained in a feature-learning and data-limited regime are affected by two ubiquitous properties of natural data: hierarchical compositionality and Zipfian distribution of features. Remarkably, the effects of these two structural properties on learning greatly differ between classification and next token prediction tasks.

For classification, we have shown that the learning curve of simple context-free grammars becomes a power law due to the broad distribution of production rules. Specifically, if production rules are power-law distributed with some exponent $a$, then the learning curve decays as $P^{-a/(a+1)}$. The exponent $a/(a+1)$ is also found in elementary toy models of memorisation of Zipf-distributed data (Hutter, 2021; Michaud et al., 2023). Yet, in our case, as in real data, this behaviour is not simply caused by memorisation, as the probability of each single datum decays exponentially with the dimension of the input. Interestingly, the pre-asymptotic phase of the learning curve can be very large and depends on the hierarchical structure of the problem.

In next-token prediction, as in classification, our analysis predicts that production rules leading to rare features require more data to be learnt and cannot be deduced from more frequent rules. Nevertheless, this effect only changes the asymptotic test loss $\mathcal{L}_\infty$ and not the scaling exponent describing how this limit is approached. The slow decay of the curve rather stems from this effect: as the training set grows, the model can resolve increasingly long-range correlations—allowing it to reconstruct deeper levels of the data's latent hierarchical structure. This suggests that the origin of scaling laws in LLMs trained for next-token prediction lies not in superficial statistics, such as feature frequency, but in the deeper hierarchical structure of the data.

## Limitations

The model of data considered is simpler than natural data, for which the topology of the tree of latent variables could vary. Furthermore, context-dependent effects may occur that cannot be described with a tree-like model. Including these properties is a promising avenue for explaining the performance differences between transformer and convolutional architectures, as well as phenomena such as in-context learning.

## Acknowledgements

The work of FC was supported by the European Union's Horizon Europe program under the Marie Skłodowska-Curie grant agreement No. 101154584. We thank Daniel,

## Impact Statement

This paper presents work whose goal is to advance the field of Machine Learning. There are many potential societal consequences of our work, none of which we feel must be specifically highlighted here

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

## A. Methods

### A.1. Deep CNNs

The deep CNNs we consider are made by stacking standard convolutional layers. To tailor the network to the structure of the data generative model, we fix both the stride and filter size of these layers to $s$. Since each layer reduces the spatial dimensionality by a factor $s$, the input size $d$ must be an integer power of $s$ and the CNNs depth equals $\log d/\log s$.

We use the Rectified Linear Unit (ReLU) $\sigma(x) = \max(0, x)$ as activation function, set the number of channels to $H$ for each layer, and consider the maximal update parametrization (Yang & Hu, 2020), where the weights are initialised as random gaussian variables with zero mean and unit variance, all the hidden layers but the last are rescaled by a factor $H^{-1/2}$, whereas the last is rescaled by $H^{-1}$. This factor causes the output at initialisation to vanish as $H$ grows, which induces representation learning even in the $H \to \infty$ limit. In practice, $H$ is set to $512$. Increasing the number of channels further does not affect any of the results presented in the paper.

Deep CNNs are trained with SGD, with the learning rate set to $H$ to compensate for the factor of $H^{-1}$. A cosine annealing scheduler reduces the learning rate by 10 over $3 \times 10^4$ training epochs. The batch size is set to the minimal size allowing convergence, where we define convergence as the training cross-entropy loss reaching a threshold value of $10^{-2}$. We use a validation set of size $2^{21}$ to select the model with the best validation loss over the training trajectory. Reported results for a given value of the RHM parameters are averaged over 10 jointly independent instances of the RHM and network initialisation for $m \leq 20$, and 3 instances for $m > 20$.

### A.2. Multi-layer self-attention (RHM)

The deep Transformers that we train on RHM data are made by stacking standard Multi-Head Attention layers (Vaswani et al., 2017), without any residuals, layer normalisation and multi-layer perceptron in between. We found that the removed components do not affect the model's performance on data generated from the RHM. Each layer has the same number of heads $n_h$ and embedding dimension $d_{\text{emb}} = n_h \times v$, with $v$ the vocabulary size. The input dimension is adapted to the embedding dimension via a learnable linear projection, to which we add learnable positional encodings. The choice of $n_h$ follows two principles: the model should be large enough for the training loss to reach a threshold value of $10^{-3}$ and changing $n_h$ should not affect performance beyond the fluctuations due to the model initialisations. Specifically, we set $n_h = 16$ and notice no significant change in performance up to $n_h = 64$. Also scaling $d_{\text{emb}}$ up to $4n_h \times v$ does not impact performance.

Multi-layer self-attention networks are trained with the Adam optimizer, with a warmup scheduler bringing the learning rate to $10^{-2}$ within the first 16 training steps. As for CNNs, the batch size is set to the lowest value allowing for convergence. We use a validation set of size $2^{17}$ to select the model with the best validation loss over the training trajectory. Reported results for a given value of the RHM parameters are averaged over 8 jointly independent instances of the RHM and network initialisation.

## B. Asymptotics of the learning curve $\varepsilon(P)$

This section contains a detailed derivation of the asymptotic learning curve for classification, Eq. 8.

Let us define $k(P)$ as the largest integer $k$ such that $f_k > vm^{L-1}/P$, corresponding to the rank of production rules that can be learnt given $P$ data according to Assumption 3.1. Substituting Zipf's law for $f_k$ yields

$$\frac{k^{-1-a}}{\sum_{j=1}^{m} j^{-1-a}} > \frac{vm^{L-1}}{P} \Rightarrow k(P) \simeq \left(\frac{P}{vm^{L-1}}\right)^{\frac{1}{1+a}}. \tag{19}$$

Let us now define $g(P)$ as the total probability of all the production rules that are learnt with $P$ data,

$$g(P) = \sum_{k=1}^{k(P)} f_k = \frac{\sum_{k=1}^{k(P)} k^{-1-a}}{\sum_{j=1}^{m} j^{-1-a}}. \tag{20}$$

Notice that, from Eq. 7, $\varepsilon(P) = 1 - g(P)^{s^{L-1}}$. To determine the asymptotic behaviour of $g(P)$, we use the Euler-Maclaurin

formula, which relates sums to integrals:

$$\sum_{k=a}^{b} f(k) = \int_a^b f(x)\, dx + \frac{f(b) - f(a)}{2} + \sum_{j=1}^{n} \frac{B_{2j}}{(2j)!} f^{(2j-1)}(x)\Big|_a^b,$$

where $B_{2j}$ are the Bernoulli numbers. In particular,

$$\sum_{k=1}^{k(P)} k^{-1-a} = \sum_{k=1}^{\infty} k^{-1-a} - \sum_{k=k(P)+1}^{\infty} k^{-1-a}$$

$$= \zeta(1+a) - \int_{k(P)}^{\infty} x^{-1-a}\, dx + \frac{k(P)^{-(1+a)}}{2} - \sum_{j=1}^{\infty} \frac{B_{2j}}{(2j)!} \frac{\Gamma(2j+a)}{\Gamma(1+a)} (k(P))^{-(2j+a)}, \quad (21)$$

where $\zeta(s)$ is the Riemann zeta function and we used that if $f(x) = x^{-(1+a)}$, then $f^{(j)}(x) = \left((-1)^j \Gamma(1 + a + j)/\Gamma(1+a)\right) x^{-(1+a+j)}$. Since $k(P)$ increases with $P$, the asymptotics of the right-hand side of Eq. 22 are controlled by the integral term, equal to $k(P)^{-a}/a$. Analogously,

$$\sum_{k=1}^{m} k^{-1-a} = \sum_{k=1}^{\infty} k^{-1-a} - \sum_{k=,+1}^{\infty} k^{-1-a}$$

$$= \zeta(1+a) - \int_{m}^{\infty} x^{-1-a}\, dx + \frac{m^{-(1+a)}}{2} - \sum_{j=1}^{\infty} \frac{B_{2j}}{(2j)!} \frac{\Gamma(2j+a)}{\Gamma(1+a)} (m)^{-(2j+a)}$$

$$\simeq \zeta(1+a) - m^{-a}/a, \quad (22)$$

hence

$$g(P) \simeq \frac{\zeta(1+a) - k(P)^{-a}/a}{\zeta(1+a) - m^{-a}/a} \simeq 1 - \frac{k(P)^{-a} - m^{-a}}{a\zeta(1+a)}. \quad (23)$$

Notice how $g(P)$ converges to 1 as $k(P)$ approaches $m$, when all production rules are learnt. For $k(P) \ll m$, instead,

$$g(P) \simeq 1 - c \left(\frac{P}{vm^{L-1}}\right)^{-\frac{a}{1+a}}, \quad (24)$$

with $c = (a\zeta(1+a))^{-1}$. By plugging this back into the equation for $\varepsilon(P)$ we get

$$\varepsilon(P) = 1 - g(P)^{s^{L-1}} \simeq 1 - \left[1 - c\left(\frac{P}{vm^{L-1}}\right)^{-\frac{a}{1+a}}\right]^{s^{L-1}}$$

$$\simeq s^{L-1} c \left(\frac{P}{vm^{L-1}}\right)^{-\frac{a}{1+a}}, \quad (25)$$

which yields the desired asymptotic behaviour in $P$.

## C. Empirical learning curves for classification with $L = 3$

This section collects additional empirical tests of the predictions of section 4 for RHM data with $L = 3$.

## D. Token-token correlations with Zipf-distributed production rules

In this section, we derive Eq. 13 for the variance of tuple-token correlations with Zipf-distributed production rules,

$$\left\langle (C_j(\boldsymbol{\mu}, \nu))^2 \right\rangle = \left(f_{k(\boldsymbol{\mu})}\right)^2 \left(\sum_{k'=1}^{m} f_{k'}^2\right) \frac{(1 - m/v^{s-1})}{v^3 m^{2\ell - 3}}. \quad (26)$$

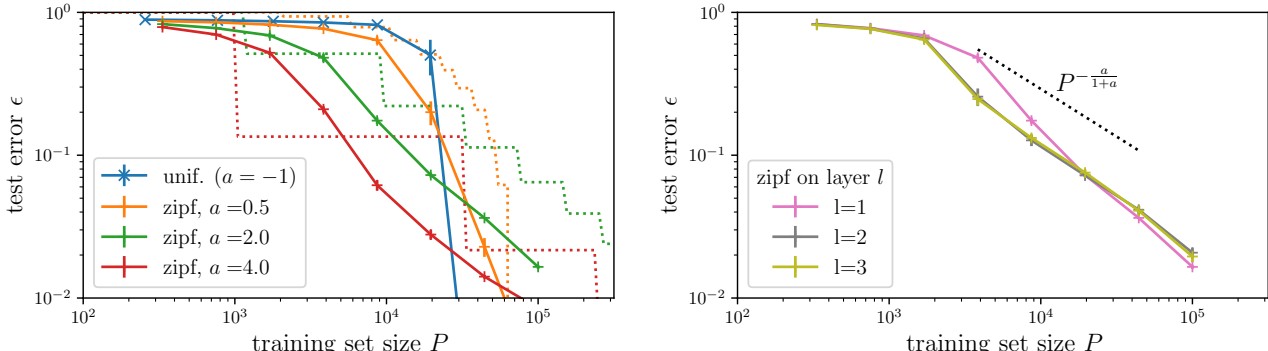

*Figure 7.* Learning Curves of classification with $L = 3$. **Left:** Learning curves of 4-layers CNNs trained on RHM data with $L = 3$, $s = 2$, $v = m = 10$ and Zipf exponent $a$ indicated in caption. Solid lines are the empirical learning curves whereas dotted lines are predictions from Eq. (7). **Right:** As in the left panel, but $a$ is fixed and the layer where production rules are Zipf-distributed changes. The black dotted line represents the scaling law $\epsilon \sim P^{-a/(1+a)}$.

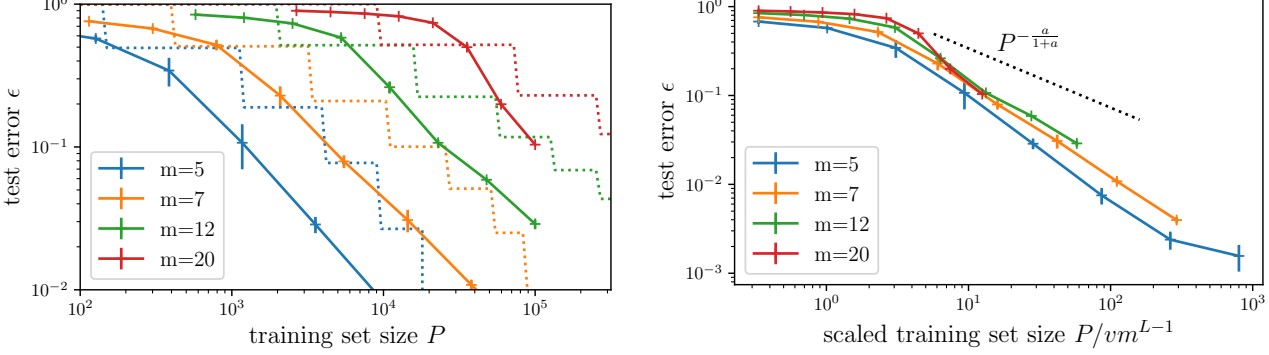

*Figure 8.* Learning curves with varying $m$ values. **Left:** Learning curves in the same setting as above, with Zipf exponent $a = 2$ and $m = v$ indicated in caption. Solid lines are the empirical learning curves whereas dotted lines are predictions from Eq. (7). **Right:** All curves collapse when rescaling the x-axis by $vm^{L-1}$—the sample complexity of an RHM with uniform production rules and $L - 1$ layers. The black dotted line represents the scaling law $\epsilon \sim P^{-a/(1+a)}$.

The starting point is the definition of the tuple-token correlation for a level-$\ell$ lowest common ancestor (compare with Eq. (124) of (Cagnetta & Wyart, 2024), Appendix F),

$$C^{(\ell)}(\boldsymbol{\mu}, \nu) = \sum_{\mu_1, \nu_1} p_{\boldsymbol{i}_1}^{(1)}(\boldsymbol{\mu}|\mu_1) p_{j_1}^{(1)}(\nu|\nu_1) C^{(\ell-1)}(\mu_1, \nu_1). \tag{27}$$

Level-1 rules are power-law distributed. The corresponding probabilities can be written as sums over production rules,

$$p_i^{(1)}(\nu|\nu_1) = \sum_{k=1}^{m} f_k \delta(r_{k,i}(\nu_1), \nu), \tag{28}$$

where the $\delta$ function equals 1 if the $i$-th symbol on the right-hand side of the $k$-th production rule coming from $\nu_1$ equals $\nu$ and 0 otherwise. $C^{(\ell-1)}(\mu_1, \nu_1)$ only involves uniformly-distributed production rules and has 0 mean, thus the mean of $C^{(\ell)}(\boldsymbol{\mu}, \nu)$ also vanishes. The variance reads, denoting with $k(\boldsymbol{\mu})$ the rank of the unique production rule generating the tuple $\boldsymbol{\mu}$, and omitting the position indices $\boldsymbol{i}_1$ and $j_1$ to ease the notation, (compare with Eq. (125) of (Cagnetta & Wyart, 2024), Appendix F)

$$\left\langle \left( C^{(\ell)}(\boldsymbol{\mu}, \nu) \right)^2 \right\rangle = (f_{k(\boldsymbol{\mu})})^2 \sum_{\nu_1, \kappa_1} \left\langle p^{(1)}(\nu|\nu_1) p^{(1)}(\nu|\kappa_1) \right\rangle \left\langle C^{(\ell-1)}(\mu_1(\boldsymbol{\mu}), \nu_1) C^{(\ell-1)}(\mu_1(\boldsymbol{\mu}), \kappa_1) \right\rangle$$

$$= (f_{k(\boldsymbol{\mu})})^2 \sum_{\nu_1} \left\langle p^{(1)}(\nu|\nu_1) p^{(1)}(\nu|\nu_1) \right\rangle \left\langle \left( C^{(\ell-1)}(\mu_1, \nu_1) \right)^2 \right\rangle$$

$$+ (f_{k(\boldsymbol{\mu})})^2 \sum_{\nu_1, \kappa_1 \neq \nu_1} \left\langle p^{(1)}(\nu|\nu_1) p^{(1)}(\nu|\kappa_1) \right\rangle \left\langle C^{(\ell-1)}(\mu_1(\boldsymbol{\mu}), \nu_1) C^{(\ell-1)}(\mu_1(\boldsymbol{\mu}), \kappa_1) \right\rangle. \tag{29}$$

By Eq. 28,

$$\left\langle p^{(1)}(\nu|\nu_1) p^{(1)}(\nu|\nu_1) \right\rangle = \sum_{k_1} (f_{k_1})^2 \left\langle \delta(r_{k_1}(\nu_1), \nu) \right\rangle$$

$$= \sum_{k_1, k_2 \neq k_1} f_{k_1} f_{k_2} \left\langle \delta(r_{k_2}(\nu_1), \nu) \delta(r_{k_2}(\nu_1), \nu) \right\rangle, \tag{30}$$

and

$$\left\langle p^{(1)}(\nu|\nu_1) p^{(1)}(\nu|\kappa_1) \right\rangle = \sum_{k_1, k_2} f_{k_1} f_{k_2} \left\langle \delta(r_{k_2}(\nu_1), \nu) \delta(r_{k_2}(\kappa_1), \nu) \right\rangle, \tag{31}$$

Using

$$\left\langle \delta(r_k(\mu_1), \mu) \right\rangle = \mathbb{P} \left\{ \mu_1 \xrightarrow{k} \mu \right\}_{RHM} = \frac{1}{v}, \tag{32}$$

where $\mathbb{P}_{RHM}$ denotes the probability over RHM realisations, and

$$\left\langle \delta(r_{k_1}(\mu_1), \mu) \delta(r_{k_2}(\nu_1), \nu) \right\rangle = \mathbb{P} \left\{ \mu_1 \xrightarrow{k_1} \mu \right\}_{RHM} \mathbb{P} \left\{ \nu_1 \xrightarrow{k_2} \nu \,\middle|\, \mu_1 \xrightarrow{k_1} \mu \right\}_{RHM}, \tag{33}$$

where

$$\mathbb{P} \left\{ \nu_1 \xrightarrow{k_1} \nu \,\middle|\, \mu_1 \xrightarrow{k_2} \mu \right\}_{RHM} = \begin{cases} \dfrac{v^{s-1} - 1}{v^s - 1}, & \text{if } \nu_1 = \mu_1, \mu = \nu, \\[2mm] \dfrac{v^{s-1} - 1}{v^s - 1}, & \text{if } \nu_1 \neq \mu_1, \mu = \nu, \end{cases} \tag{34}$$

we get

$$\left\langle p^{(1)}(\nu|\nu_1) p^{(1)}(\nu|\nu_1) \right\rangle = \left( \sum_k f_k^2 \right) \frac{1}{v} + \left( 1 - \sum_k f_k^2 \right) \frac{1}{v} \frac{v^{s-1} - 1}{v^s - 1} \tag{35}$$

and

$$\left\langle p^{(1)}(\nu|\nu_1)p^{(1)}(\nu|\kappa_1) \right\rangle = \frac{1}{v}\frac{v^{s-1}-1}{v-1}, \tag{36}$$

In addition, the covariance of the correlations satisfies (Eq. (74) of (Cagnetta & Wyart, 2024))

$$\sum_{\kappa \neq \nu} \langle C(\mu,\nu)C(\mu,\kappa) \rangle = -\left\langle C(\mu,\nu)^2 \right\rangle. \tag{37}$$

Therefore,

$$\left\langle \left(C^{(\ell)}(\boldsymbol{\mu},\nu)\right)^2 \right\rangle = (f_k(\boldsymbol{\mu}))^2\, v \left[ \left(\sum_k f_k^2\right)\frac{1}{v} + \left(1 - \sum_k f_k^2\right)\frac{1}{v}\frac{v^{s-1}-1}{v^s-1} - \frac{1}{v}\frac{v^{s-1}-1}{v-1} \right] \left\langle \left(C^{(\ell-1)}(\mu_1,\nu_1)\right)^2 \right\rangle$$

$$= (f_k(\boldsymbol{\mu}))^2 \left(\sum_k f_k^2\right)\frac{v^s}{v^s-1}\frac{v-1}{v}\left\langle \left(C^{(\ell-1)}(\mu_1,\nu_1)\right)^2 \right\rangle, \tag{38}$$

which, after substituting the correlations of the uniform case from (Cagnetta & Wyart, 2024), yields Eq. 13 in the limit of large $v$.

## E. Cross-entropy estimate in the limit $1 \ll m \ll v^{s-1}$

Here we derive the approximation of $s^\ell$-gram cross-entropies Eq. 18. Let us start from Eq. 17 of the main,

$$\langle |\mathcal{V}_{c,\ell}| \rangle = \frac{1}{1 - m/v^{s-1}} + v\left(\frac{m}{v^{s-1}}\right)^{\ell-1}. \tag{39}$$

for the average size of the set of level-1 tokens compatible with the $s^\ell$-gram context. In the limit where $1 \ll m \ll v^{s-1}$, $\langle |\mathcal{V}_{c,\ell}| \rangle$ approaches 1 as $\ell$ grows. In this limit, we can neglect the probability that $|\mathcal{V}_{c,\ell}| > 2$, i.e.

$$\langle |\mathcal{V}_{c,\ell}| \rangle = \sum_{n=1}^{v} n\mathbb{P}\left\{|\mathcal{V}_{c,\ell}| = n\right\} \simeq q_1 + 2q_2, \tag{40}$$

where

$$\begin{cases} q_1 := \mathbb{P}\left\{|\mathcal{V}_{c,\ell}| = 1\right\} = 2 - \langle |\mathcal{V}_{c,\ell}| \rangle, \\ q_2 := \mathbb{P}\left\{|\mathcal{V}_{c,\ell}| = 2\right\} = \langle |\mathcal{V}_{c,\ell}| \rangle - 1. \end{cases} \tag{41}$$

Accordingly, we split the RHM data $\boldsymbol{x}$ into those with $|\mathcal{V}_{c,\ell}| = 1$ (fraction $q_1$) and those with $|\mathcal{V}_{c,\ell}| = 2$ (fraction $q_2$).

If $|\mathcal{V}_{c,\ell}| = 1$, then $\mu^{(1)}$ can only attain the unique symbol $\bar{\mu}^{(1)}$ that generates the $s$-tuple $(x_{d-(s-1)}, \ldots, x_{d-1}, x_d)$. However, there could be other production rules generating $(x_{d-(s-1)}, \ldots, x_{d-1}, \tilde{x})$ from $\bar{\mu}^{(1)}$, yielding a positive contribution to the entropy. For each of the $\tilde{x} \in \mathcal{V}_0$, the probability that such production rule exists is $(m-1)/(v^s - 1)$. Since there are $v-1$ elements of $\mathcal{V}_0$ other than $x_d$, the total probability is $(v-1)(m-1)/(v^s-1) \simeq m/v^{s-1}$. In the limit $1 \ll m \ll v^{s-1}$ this probability is small, thus, as for the level-1 symbol $\mu^{((1)}$, we neglect the possibility that there is more than one $\tilde{x}$ compatible with $(x_{d-(s-1)}, \ldots, x_{d-1}; \bar{\mu}^{(1)})$. To sum up, there is a fraction $q_1(m/v^{s-1})$ of data having $|\mathcal{V}_{c,\ell}| = 1$ but two distinct tokens $\tilde{x}$ and $x_d$ compatible with the context $(x_{d-(s-1)}, \ldots, x_{d-1}; \bar{\mu}^{(1)})$.

When $|\mathcal{V}_{c,\ell}| = 2$, $\mu^{(1)}$ attains $\bar{\mu}^{(1)}$ or $\tilde{\mu}^{(1)}$. One of the production rules emanating from $\tilde{\mu}^{(1)}$ could be compatible with $(x_{d-(s-1)}, \ldots, x_{d-1})$, yielding a positive contribution to the cross-entropy. The probability that this production rule exists is again $m/v^{s-1}$ and we neglect the possibility of two such rules as $1 \ll m \ll v^{s-1}$. Moreover, even if such a production rule does not exist, there could be, as in the case $|\mathcal{V}_{c,\ell}| = 1$, another $\tilde{x}$ compatible with $(x_{d-(s-1)}, \ldots, x_{d-1}; \bar{\mu}^{(1)})$. The resulting fraction of data with positive entropy is $2q_2(m/v^{s-1})$, yielding a total fraction $(q_1 + 2q_2)(m/v^{s-1}) = \langle |\mathcal{V}_{c,\ell}| \rangle (m/v^{s-1})$ of contexts compatible with 2 entries of $x_d$. The entropy contribution of each of these contexts is the logarithmic sum of two probabilities taken from the set $(f_k)$ of production rules probabilities. The average contribution over the possible choices of two $f_k$'s is independent of $\ell$. Denoting this average with $H_{2,a,m}$, and using Eq. 17 for $\langle |\mathcal{V}_{c,\ell}| \rangle$, we get Eq. 18 of the main paper,

$$\mathcal{L}_\ell \xrightarrow[1 \ll m \ll v^{s-1}]{} H_{2,a,m}\left(\frac{m/v^{s-1}}{1 - m/v^{s-1}} + v\left(\frac{m}{v^{s-1}}\right)^\ell\right). \tag{42}$$