# OpenReview forum: "Learning curves theory for hierarchically compositional data with power-law distributed features"
_ICML.cc/2025/Conference — ICML 2025 poster_

### Official Review · Reviewer_xKcS · 2025-03-10

**Overall Recommendation:** 3

**Summary:**

This paper uses a theoretical framework to explain the emergence of neural scaling laws. It builds on the synthetic model and results derived in previous works on classification (Cagnetta et al., 2024) and next-token prediction (NTP) (Cagnetta & Wyart, 2024) that studied how hierarchical tasks are learned. The authors use the Random Hierarchy Model (RHM), a type of probabilistic context-free grammar, to generate synthetic data with a controllable hierarchical structure. This allows them to analyze how the loss changes as the training dataset grows in size.

The previous two works focused on the case where the hierarchical rules generating the data were uniformly distributed. This paper extends those analyses to the case where the rule probabilities follow a power-law distribution (at a single level of the hierarchy). In the case of classification tasks, the authors show that the scaling law exponent depends on the rule distribution. However, for next-token prediction, they argue that the exponent remains unchanged across different power-law distributions. While the decay rate of the loss remains the same regardless of the exponent, the rule distribution affects the final loss value, which corresponds to the entropy lower bound. They support their claims with both theoretical analysis and experiments on RHM models.

**Claims And Evidence:**

The derivation of their theoretical results relies on an assumption (Assumption 3.1), which is not rigorously proven but is intuitively used. This assumption states that as long as the correlation between features is distinguishable from noise, the model can correctly predict the class or next token. Based on this assumption, the authors, without proof, use the fact that if the correlations are statistically resolvable, the model can always correctly classify the sequences in the case of classification, or that the cross-entropy loss matches the loss of an $\ell$-gram model in the case of next-token prediction (a similar assumption is made in Cagnetta & Wyart (2024)).

Apart from this assumption, the rest of the theory is solid and extends the asymptotic analysis from the previous works by Cagnetta & Wyart (2024) and Cagnetta et al. (2024) to the case where the rules have non-uniform distribution.

The empirical results validate the accuracy of this asymptotic analysis for CNNs/Transformers trained on the RHM data.

**Essential References Not Discussed:**

None that I’m aware of.

**Experimental Designs Or Analyses:**

The experimental analysis fits within the scope of the paper and supports its claims.

(See "Weaknesses" for further comments on the experiments.)

**Methods And Evaluation Criteria:**

The theory and empirical results support the scope and claim of the paper for the RHM setup.

**Other Comments Or Suggestions:**

Minor typos:
- line 217 first column, Theorem in "Assumption Theorem 3.1" is extra.
- Line 159, second column, have you defined $n_c$ in the expression for $P^*(\mathbf{\mu})$?
- From what I understand, the analysis is for the joint expectation of the loss over the realization of RHM and the sequences $x$. Is that also the case in eq (14) (and (15))? The current notation used in the equation only takes expectation over the sequences.

**Other Strengths And Weaknesses:**

**Strengths**:

The synthetic setup is nice: compared to previous papers that focus on regression tasks to derive scaling laws, the RHM provides a closer model to language, where the problem consists of discrete vocabularies and sequential data. While it cannot perfectly capture all aspects of natural language, it offers a nice, controllable framework that models the hierarchical and tokenized nature of language.

**Weaknesses**:

1. One aspect that needs to be emphasized in the main body is the training setup. Scaling laws (e.g, as in Kaplan et al. (2020)) can be derived in different setups depending on whether data, model size, or FLOPs is the bottleneck.

    I assume in this paper, data is the bottleneck, while model size and FLOPs are flexible (can be unbounded). It's mentioned in App A. that the classification experiments are run for a large number of epochs. However, this detail seems to be missing for the next-token prediction experiments. Whether the model is trained for a single pass or multiple epochs, and the size of the model, might influence the behavior of the scaling laws. Specifically, it would be useful to know whether the training length was long enough and the model size large enough for the training loss to reach its lower bound, or if the experiments were conducted in the under-trained regime. Additionally, it would be helpful to know whether the test loss for each given dataset size $P$ has converged.

2. The emergence of scaling laws for NTP in the case of RHM, due to the hierarchical structure of the data, was already demonstrated in Cagnetta & Wyart (2024). In that sense, this paper does not offer additional insights for NTP. However, in the case of classification, the observation that changing the distribution of the rules can alter the scaling behavior is an interesting observation.

**Questions For Authors:**

In Sec 3.1, the different loss behavior in the case of classification is attributed to the difficulty of resolving the correlations for the rare production rules. A few questions regarding this argument:
1. In the figures, the larger $a$ is (further away from the uniform distribution), the error/loss decays faster in both cases. So, is having the Zipf's distribution over the rules making the task easier?
2. The same argument about the rare rules and correlation should hold true in the case of NTP as well. But why isn't the behaviour changing?
3. Given that your analysis shows the scaling behavior is unchanged, can you explain the source of the gap between the loss curves for different values of $a$ in the NTP task? Based on the analysis, the only thing that should change with $a$ is the cross-entropy lower bound, which is not surprising since changes in the distribution affect the entropy of the data.
4. Can you explain what happens with $a=-1$ (uniform rule distribution) in the classification case? Eq (7) cannot be evaluated at this value.

**Minor question:**

5. In Equation (6), is the right-hand side (RHS) an equality, or is it only an upper bound on the error? The RHS measures when all patches are resolvable, but a model could still make a correct prediction by resolving the majority of patches correctly, right?

**Relation To Broader Scientific Literature:**

There is a substantial body of empirical papers on scaling laws. As also listed in the paper, many previous works explain the scaling laws from a theoretical perspective in simplified setups like linear/kernel regression.

This paper builds on the previous works by Cagnetta & Wyart (2024) and Cagnetta et al. (2024), which use the RHM to study learning from hierarchical data. These earlier works derived loss behavior and scaling laws under the assumption of a uniform rule distribution. This paper extends that analysis to consider non-uniform rule distributions.

**Theoretical Claims:**

(I followed the proof sketch from the main body but skipped the details in the supplementary material. See "Claims And Evidence".)

---

> ### Author Rebuttal · Authors · 2025-03-25
>
> ## Weaknesses
>
> 1. As the reviewer points out, we work in the data-limited regime, which is suitable for studying learning curves. In all our experiments, the models are large enough and trained for long enough for the training loss to reach its lower bound. In particular, for consistency, we increase the number of parameters (width for CNNs and number of heads for transformers) until the performance does not depend on this number anymore. Notice that, being overparametrized, the test loss of our models increases towards the end of training due to overfitting. To solve this problem, we employ early stopping, i.e. select the training step where the model gave the best performance over a validation set of size between ~32K and ~131K (depending on the Zipf exponent $a$). Studying how many parameters are needed is interesting, but we view it as less fundamental than the present work. The number of parameters will presumably depend on the detailed choice of architecture (transformers vs CNNs, depth vs width etc.). In contrast, we expect our present results to hold for deep enough architectures, including CNNs and transformers (as shown in the absence of Zipf distribution in Cagnetta et al, 2024). ***In the revised manuscript***, we will emphasize further that our results apply to the data-limited regime, and comment in the conclusions about the possibility of studying the dependence on FLOPs and model size.
>
> 2. We think that the fact that the scaling law of NTP does not depend on the Zipf law is a surprising and interesting result. This result is particularly relevant for a whole portion of the theoretical literature on scaling laws, assuming that scaling results from the Zipf distribution of the data features. In particular, our result suggests that the approach based on the feature distribution is not the right one for NPT, although it may be suitable for classification.
>
> ## Questions
>
> 1. Rather than making the task easier, having Zipf-distributed production rules results in the separation of data into different categories. Those having only the most frequent production rules ($k=1,2,\dots$) are easier to learn compared to the uniform case $f_k=1/m$, whereas those having some rare production rules $k=m,m-1,\dots$ are harder. As a result, the sigmoidal learning curve of the uniform case turns into a power law. ***In the revised manuscript***, we will add the learning curves of classification in the uniform case to Figures 2 and 7 to clarify this point.
>
> 2. The behaviour of the individual steps is indeed changing, as highlighted in Figure 4. However, when considering a sequence of steps, the overall scaling is determined by the distance between the steps (which is controlled by the hierarchical structure) rather than the behaviour of the single step (controlled by the ZIpf law).
>
> 3. We should stress that our result is asymptotic, so it does not apply to the very first step. That said, what changes with $a$ is not only the cross-entropy lower bound (which causes a vertical shift of the curves), but also the sample complexities of the steps from Eq. (13) (causing a horizontal shift of the curves). However,  the way that such sample complexities depend on the level of the step is independent of $a$: that's why the scaling law remains unchanged.
>
> 4. In the uniform case $f_k=1/m$ for all $k$'s. There is only one sample complexity $P^*=v m^L$, and the test error crosses over from $\simeq 1$ to $\simeq 0$ around $P^*$. Following the suggestion of Reviewer gXnS, we will add further background material on Cagnetta et al., 2024, that clarifies this point.
>
> 5. The reviewer is correct that it is an upper bound. Depending on the value of $m/v^{s-1}$, the root can still be inferred correctly without resolving all of the patches. The probability for this to happen is derived in (Sclocchi et al., PNAS 2024) for an optimal decoder. However, notice that, if $m=v^{s-1}$ as in figures 2 and 3, changing one patch changes the root label with high probability, implying that all patches need to be resolved. ***In the revised manuscript***, we will add a sentence after equation (6) to clarify this point.

---

> > ### Comment · Reviewer_xKcS · 2025-04-04
> >
> > Thanks for the clarification.
> >
> > I’m increasing my score to 3. The discussion in the rebuttal would be a valuable addition to the paper, particularly in clarifying the exact setup for the scaling law and making the discussions more self-contained, as suggested by Reviewer gXnS.
> >
> > The setup and methodology rely heavily on previous work (Cagnetta & Wyart, 2024; Cagnetta et al., 2024), making the contribution somewhat incremental. However, the results are sound and relevant to ongoing theoretical discussions. The contrast between classification and NTP is interesting, and studying the impact of long-tailed feature distributions in these setups could be useful to the community.

---

### Official Review · Reviewer_gXnS · 2025-03-13

**Overall Recommendation:** 2

**Summary:**

- Main algorithmic/conceptual ideas:
  - Mostly based on the framework of Cagnetta et al., 2024.
- Main results/findings:
  - This paper shows that Zipf distribution of feature exists (which is common in real world scenario for long-tailed distributions) the learning curve of classification will show a power law shape instead of a sigmodal shape during uniform distribution.
  - The curve of next-token prediction won't change due to Zipf assumption.

**Claims And Evidence:**

Empirical results support all theoretical claims, which is good. For the discussion of theoretical claims, see relation to broader scientific literature section.

**Essential References Not Discussed:**

No I'm not aware of any missing essential references.

**Experimental Designs Or Analyses:**

Experiments mostly look good to me. One question: for Figure 6, what is the black dashed line? Why is this black dashed line a better prediction for top three colored curves compared to the red dashed line representing Eq. 11?

**Methods And Evaluation Criteria:**

- Methods: N/A because this is a theory paper.
- Evaluation Criteria: The scaling law is verified with synthetic dataset. Since it's designed for the purpose of verifying theoretical statements, this is acceptable.

**Other Comments Or Suggestions:**

- Line 125: It's better to clarify $s$ represents a binary tree and $L$ is the height of the root node $-1$ if my understanding is correct.
- Line 339: replace the notation of normalization as $\mathcal{N}$ is often used to describe Gaussian distribution.
- Comment on writing: It will be better to add one preliminary section before section 3 to incorporate all materials before nonuniform scenarios, because these are all background materials. The current sections are a bit confusing: when I read the paragraph starting from line 110, it's hard for me to tell if this is from your contribution or from Cagnetta et al., 2024 which you only cite in the early sentences of the paragraph. Or alternatively, use one sentence before section 3 to tell the readers what to expect in the following section.

**Other Strengths And Weaknesses:**

- Originality: This paper extended Cagnetta et al., 2024 and I believe this is a novel contribution. The motivation is clear, although I'm not sure how is your specific choice to inject Zipf distribution related to the practical problem you want to solve. See the question for Line 70.

- Significance:
  - Given the limitation authors also realized that their HRM model might be still a simplification of the real-world data, I do want to ask how are your conclusions guide practitioners.
  - We see all proved scaling laws are verified empirically so I don't have too much concerns about the correctness of the statements. And this is a big strength for this paper.

- Clarity: have some space for improvement. See Relation To Broader Scientific Literature. Also, I noticed that you do have the space at the end of the 8th page so it's better to elaborate more on background papers. I would recommend the author to make sure all knowledge required to understand this paper is self-contained.

**Questions For Authors:**

See all comments above. The current score is mainly decided based on the writing issue, but this could be addressed in the rebuttal session if the authors can provide convincing explanations for questions about theoretical technical details.

**Relation To Broader Scientific Literature:**

- Based on the whole paper, it seems that this paper is mainly built on two papers, Cagnetta et al., 2024 (learning curve theory based on RHM) and Hutter, 2021 (standard learning curve theory).
- However, I feel the current description of these two papers are insufficient to let the reader understand the current submission without checking these two background papers carefully.
- For example, more background for these papers can be very helpful to solve the following questions:
  - How did you derive Eq. (4)?
  - Line 147: where does the quadratic term of $f_{k(\mu)}$ come from?
  - Line 155: Why do you need to divide $P$ by $f_{k(\mu)}m$?
  - Line 192: Could you elaborate on how to compute two mentioned correlations with Cagnetta et al., 2024? What claims in Cagnetta et al., 2024 are used?
  - Line 262: What's the framework in Hutter, 2021 specifically and the relationship to your paper?
  - Equation 13: This seems to be from rearranging some terms in Eq.12, but how do you derive Eq.13 exactly and what's the meaning of $P_{l,k}$?

**Theoretical Claims:**

Yes, for all claims made in the main text. The writing of the paper makes it a bit hard to judge theoretical claims made in the paper. Also see relation to broader scientific literature section.

A couple of questions about the theory part:
- What does "compatible" mean? Line 86 it says: "training data is compatible with both a PCFG and a nonhierarchical generative model", Line 343 it says "the set of terminal tokens compatible with the context of size ..." and there are 13 mentions of "compatibility" in this submission. They don't sound like sharing the same meaning. What is "compatible" with what?
- Line 70: Even if power law exists everywhere in practice, what does it imply to model Zipf for each layer in your tree?
- Line 130: "In this case,... last token $x_d$" which 2-tuples on the figure are considered in the next-token prediction task?
- Line 132: Why does $f_{k(\mu)}$ depend on the rank (in your subscript, also inferred from line 148 and no information is found before line 132)? Also, what are you ranking exactly? Is it related to line 161 when you wrote "Ranking all the low-level tuples by the probability of the corresponding production rules" later on?

---

> ### Author Rebuttal · Authors · 2025-03-25
>
> ## Theoretical claims
>
> 1. **compatible**. In line 86, compatible means that the sentences in the training data can be generated both by a PCFG and by a non-hierarchical generative model. The next 4 occurrences of "compatible" in the main refer to compatibility of a single token ($x_d$) with the context ($x_1,\dots,x_{d-1}$), where compatibility is defined by the full string ($x_1,\dots,x_d$) belonging to the set of strings generated by the RHM. The next two occurrences refer to a different kind of compatibility, i.e. the one of an empirical curve that follows a theoretical prediction within the error bars. ***In the revised manuscript***, we will define the compatibility of a token with the context as above, and we will use "agrees with the theory" instead of "is compatible with the theory" when referring to empirical curves.
>
> 2. **Zipf for each layer**. Having Zipf production rules at all levels would change the distribution of the input features, as their probability will be given by the product of several Zipf-distributed frequencies. Nevertheless, the learning curve can still be described via Assumption 3.1. In particular, we believe that the main conclusion of our paper would still hold: the exponent of the scaling law is controlled by the Zipf distribution of production rules for classification and by the hierarchical structure for next-token prediction.
>
> 3. **Which $2$-tuples**. All the complete $2$-tuples are considered: $(x_5,x_6)$ for reconstructing $\mu_3^{1}$, $(x_3,x_4)$ for $\mu_2^{1}$ and $(x_1,x_2)$ for $\mu_1^{1}$. However, unlike the classification case, the correlations are not equal for all tuples: $(x_5,x_6)$ is closer in tree distance to the missing token, so it has a higher correlation. As a result, the hidden variable $\mu_3^{1}$ can be inferred with less samples than $\mu_1^{1}$ and $\mu_2^{1}$.
>
> 4. **Dependency on the rank**. The rank $k$ referred to here was introduced in assumption *iv)* of Section 2, $k\,{=}\,1,\dots,m$ from the most to the last likely production rule. Due to the non-ambiguity assumption, the rank determines uniquely the low-level tuple $\mathbf{\mu}$ produced, thus $k=k(\mathbf{\mu})$. This is indeed connected to the sentence "Ranking all the low-level tuples by the probability of the corresponding production rules". ***In the revised manuscript***, we will clarify this point while discussing the derivation of Eq. (4) (see relation to literature paragraph).
>
> ## Experimental designs
>
> The black dashed line in Figure 6 represents the prediction in the uniform case and should only describe the top coloured curve (solid blue line). However, as we claim in the paper, all curves display the same asymptotic decay with $P$, as highlighted by the red dashed curve. This curve is only supposed to represent the asymptotic power-law decay emerging from the sequence of steps. Its absolute position has been shifted by adding a $P$-independent constant for visualisation purposes.
>
> ## Relation to broader literature
>
> As the reviewer suggests, ***in the revised manuscript***, we will add a section before 3, including,
>
> 1. Further background on Cagnetta et al., 2024, in particular concerning the calculation of correlations joint probabilities like Eq. (4) using the CFG structure, and the derivation of the sample complexities required for the accurate measurement of said correlations. In simple terms, the joint probability of the leaves conditioned on the class is given by the product of the probabilities of all the production rules used. Then, correlations involving a specific tuple of leaves or a specific production rule are obtained by marginalisation. The corresponding sample complexity is derived by studying how the empirical correlations measured with $P$ training data converge to the true correlations as $P$ increases. This addition would solve questions about Eq. (4), line 147, line 155, line 192 and Eq. (13).
>
> 2. Further background on Hutter, 2021. Hutter ranks all the input data according to their probability, and assumes that a datum is correctly classified only if it appears in the training set (memorisation). We, instead, rank data according to the probability of the associated production rules, and assume that data can be correctly classified when all these production rules can be resolved.
>
> ## Significance for practitioners
>
> Despite the simplicity of the data model, our results can be directly tested in real scenarios, for instance, by training LLMs on real data where the rarest tokens/words/grammatical constructions have been removed. Such empirical studies could suggest practical directions to explore, for example, curriculum learning approaches where the fraction of rare words or grammatical constructions would be varied during training. For RHM data, such a procedure will not alter the learning curve exponent but may change the prefactor.
>
> ## Other comments
>
> We will clarify line 125 and replace $\mathcal{N}$ in line 339.

---

> > ### Comment · Reviewer_gXnS · 2025-04-04
> >
> > Thanks for authors' rebuttal! All my questions for technical details were answered, but
> > 1. since the work is heavily based on previous work, it's hard for me to justify the relative significance of your contribution, even if it's novel.
> > 2. rewriting to include further background as you listed requires a nontrivial amount of work
> > 3. Significance for practitioners is not proved
> >
> > Therefore, I hold the current score. Feel free to leave comments and I'm open to discussion.

---

> > > ### Author Response · Authors · 2025-04-05
> > >
> > > We are surprised by the reviewer’s answer, who agrees with all our technical points but maintains a mark below acceptance based on generic comments.
> > >
> > > **Concerning 1**,  our main results explaining how rare tokens or grammatical rules affect learning curves are entirely new. In particular, our result on next-token prediction questions many other theoretical works assuming that scaling results from the ZIpf distribution of the data features (see reply to Reviewer xKcs). Considering how scaling laws have driven the technological advancement of LLMs,  it is key to understand what controls scaling exponents. The significance of our results is thus clear, as agreed by all other reviewers.
> > >
> > > **Concerning 2**, the additional background can be integrated efficiently without significant effort, as it comes from already published material, and will enhance the manuscript clarity
> > >
> > > ***Concerning 3***, ICML not only publishes practical works with proven applications, but also fundamental works that help the community as a whole establish understanding, which can lead to new technology on a longer time scale.
> > >
> > > As a result, we believe that there are no specific scientific reasons that justify the below-acceptance mark.

---

### Official Review · Reviewer_1eLH · 2025-03-16

**Overall Recommendation:** 4

**Summary:**

This paper presents a theoretical model to explain the emergence of power-law learning curves in deep neural networks trained on data with Zipfian feature distributions and hierarchical compositional structure. By parameterizing the Random Hierarchy Model (RHM) with a probabilistic context-free grammar (PCFG), the authors obtain asymptotic scaling laws for classification and next-token prediction tasks. For classification, they show that if the production rules satisfy a Zipf law with exponent a, then test error asymptotically decreases as ε(P) ∼ P^(–a⁄(1+a)) (to within a large multiplicative constant depending on the combinatorial structure). For next-token prediction, while the fine-grained shape of the representation learning curve depends on the production rule distribution, large-scale scaling is only dictated by the hierarchical form. These theoretical results are verified by large-scale numerical experiments on deep convolutional neural networks and transformers on artificial data from the RHM.

## Update after rebuttal
I stay with my original rating, as I agree with the comments of author's rebuttal.

**Claims And Evidence:**

Its main arguments are two-pronged. One, the paper states that in classification tasks the learning curve attains the Zipf exponent, giving a power-law decay with exponent a⁄(1+a) after a pre-asymptotic phase ruled by the hierarchy. Second, for next-token prediction tasks the asymptotic decay is independent of the Zipf exponent, with the hierarchical structure being the primary determinant of the scaling behavior. These claims are supported by asymptotic derivations (see Equations 6–7 and 11 in the paper) and are validated through systematic experiments (illustrated in Figures 2–7) that show agreement between theory and empirical learning curves. Overall, the evidence seems mathematically rigorous and empirically convincing, though the experiments are conducted solely on synthetic data.

**Essential References Not Discussed:**

All essential references seem to be discussed.

**Experimental Designs Or Analyses:**

Experiments are performed to confirm the theoretical predictions in controlled settings. The authors generate synthetic data with the RHM for varying values of the Zipf exponent and the number of production rules. They then train deep CNNs and transformers, reporting both classification error and cross-entropy loss. The experimental design effectively isolates the role of the hierarchical structure and the nonuniform (Zipf) distribution. The empirical learning curves—especially their collapse after rescaling—strongly support the theoretical analysis. One potential concern perhaps is that the experiments are limited to synthetic data; additional validation on real-world datasets might further strengthen the findings.

**Methods And Evaluation Criteria:**

They derive their scaling laws by considering the statistics of correlations in RHM-generated data. They assume that a production rule is "learned" when an effect of a production rule on correlations is seen (Assumption 3.1) and use it to infer sample complexities at different levels of the hierarchy. In experiments, the paper employs deep CNNs for classification and multi-layer transformers for next token prediction, and tracks test error and cross-entropy loss as functions of training set size. The test criteria—i.e., the asymptotic decay of loss and error, and the collapse of rescaled learning curves—are appropriate for demonstrating the predicted power-law behavior.

**Other Comments Or Suggestions:**

.

**Other Strengths And Weaknesses:**

Strengths:
The paper provides a good theoretical framework that nicely bridges the hierarchical organization of data and power-law learning curves.
Derivations are mathematically accurate and relatively comprehensive.
Experimental evidence on synthetic data is in good agreement with theory predictions.
The paper provides useful insights which can potentially explain the behavior of large-scale deep learning models.

Weaknesses:
The reliance on a model of synthetic data (RHM) prevents direct extension to real data.
Assumption 3.1, while calculatively convenient, may be to idealized and its behavior under more realistic conditions may not be explored sufficiently.
Additional experiments with natural datasets would strengthen the contribution.

**Questions For Authors:**

.

**Relation To Broader Scientific Literature:**

The paper situates its contribution in the broader framework of neural scaling laws and theoretical knowledge of deep learning. The paper offers a new perspective on why power law scaling is achieved in deep learning. The connection to both empirical and theoretical work is made explicit throughout the manuscript.

**Theoretical Claims:**

The paper presents several theoretical claims, including the derivation of the sample complexity P* that scales as vm^L, and the asymptotic decay ε(P) ∼ P^(–a⁄(1+a)) for classification. The derivations are detailed and build on top of previous work on neural scaling laws and learning curve theory. Although the derivations make strong assumptions (e.g., Assumption 3.1), the mathematical arguments appear sound and well-connected to prior results in the literature. The paper also formally derives the scaling law for next-token prediction (Equation 11) and describes how the hierarchical structure allows deep models to capture increasingly longer-range correlations.

---

> ### Author Rebuttal · Authors · 2025-03-25
>
> We thank the reviewer for appreciating our study's strengths. Regarding the connection with real data, please see the reply to Reviewer GeDB.
>
> Concerning experiments on natural datasets, our work predicts that removing rare words from a data set should not affect the scaling law of the learning curve. It also predicts that grammatical structures that involve a large number of tokens require more data to be learned. Studying these questions systematically during training, beyond the existing analyses of how pre-trained LLMs represent grammar, will require a whole new project. We believe that presenting the theory clearly and convincingly warrants a dedicated paper, as we do here.

---

### Official Review · Reviewer_GeDB · 2025-03-17

**Overall Recommendation:** 3

**Summary:**

This paper extends the findings in (Cagnetta et al., 2024) to a probabilistic context-free grammars (PCFGs) case. The authors investigate how the structure of data influences learning curves, focusing on datasets with hierarchical compositional structures and features distributed according to power laws (Zipf distributions). By using probabilistic context-free grammars (PCFGs) to generate synthetic data, the authors unify two theoretical perspectives explaining neural scaling laws.
- For classification tasks, a Zipf distribution of features transforms learning curves from sigmoidal to power-law shapes.
- For next-token prediction tasks, instead, the hierarchical structure primarily dictates the power-law scaling exponent, while the Zipf distribution.

## update after rebuttal
I stay with the original rating.
The merits of the theoretical contribution made me incline to weak acceptance.
The common concerns about its applicability to real-world data (also mentioned by Reviewer 1eLH) and applications stopped me from further increasing my rating.

**Claims And Evidence:**

Two major claims are supported by both theoretical analysis and experimental results on synthetic data.

**Essential References Not Discussed:**

N/A

**Experimental Designs Or Analyses:**

The experimental design is a natural extension of (Cagnetta et al., 2024), which looks sound and valid. one concern would be the model size and data size are too small in claiming ``scaling law``.

**Methods And Evaluation Criteria:**

The evaluation is based on the theoretical framework proposed in the paper. The experiments are mainly designed to verify this theoretical framework, which makes the data modeling depending on certain assumptions, e.g., the PCFGs assumption. But it makes sense as the observation of ''leaf-level nodes following Zipf's law'' has been observed in textual data.

**Other Comments Or Suggestions:**

N/A

**Other Strengths And Weaknesses:**

One potential weakness would be the application power of the proposed scaling law. The results (both theoretical and experimental) rely on the assumptions of PCFGs, which is not necessarily true for the data we observed in today's applications. I further raise questions in the ``Question for Authors`` section.

**Questions For Authors:**

How far are the results proposed by the authors from guiding the applications like LLMs? This question can be further divided into the following parts:
- What is the exact size of the model proposed in the paper? The description in Appendix A refers to the model size implicitly using $H$ for CNNs or $n_h$ for self-attentions, but the exact size of the model is not reported.
- The proposed synthetic dataset is relatively small, at the scale of $\sim 10^6$,  is this enough for claiming "the scalling law" as today's scaling law usually refers to much larger data sizes?
- How different the synthetic data from the natural data? As mentioned in (Tomasini & Wyart, 2024), a context-free grammar modeled data is applicable for hierarchical data, but how about the case of natural data? References or comparisons with textual, vision, and audio would be appreciated.

**Relation To Broader Scientific Literature:**

The paper contains proper related literature, most prominently (Cagnetta et al., 2024).

**Theoretical Claims:**

I checked Eq. (7) and Eq. (12), corresponding to two major claims of this paper, their proofs in Appendix B and D are correct from my understanding.

---

> ### Author Rebuttal · Authors · 2025-03-25
>
> **1st question:** For deep CNNs, $H=512$ (1M parameters in total, see last sentence of the second paragraph in appendix A), and we checked that our conclusions do not change up to $H=1024$ (~4M parameters). For transformers, $n_h=16$ (resulting in ~3M parameters), and we checked that our conclusions do not change up to $n_h=64$. In simple terms, we set the number of parameters to be large enough that it does not impact performance. As a result, we can focus our study on the dependency on the size of the training set (see also reply to Reviewer xKcS). We will clarify the text accordingly.
>
> **2nd question:** Our predictions in the synthetic data setting are valid for arbitrarily large datasets. Fixing the RHM parameters (in particular $v$, $m$ and $L$) to the values considered in the paper allows us to test the theory with $\leq 10^7$ data, which is the range accessible to our experiments.
>
> **3rd question:** Concerning language, CFGs are believed to provide an accurate representation of syntax, except for only a few languages and constructions (see e.g. Gerhard Jäger and James Rogers, *Formal language theory: refining the Chomsky hierarchy*, Philosophical Transactions of the Royal Society B: Biological Sciences, 367(1598):1956–1970, 2012). Some aspects of semantics can also be modelled with trees via "attribute grammars" (Knuth, *Semantics of context-free languages*, Mathematical systems theory 2.2 (1968): 127-145), where hidden non-terminal symbols are complemented by attributes that influence the choice of production rule and are transmitted along the tree. Natural images can also be described with stochastic grammars, as done in pattern theory (Ulf Grenander, *Elements of pattern theory*, JHU Press, 1996). The RHM considered in this paper makes several simplifying assumptions to be tractable: uniform production rule probabilities,  frozen tree topology and random production rules. Despite the approximations, the RHM already captures non-trivial phenomena observed in real data, as shown in (Cagnetta et al. 2024) and (Sclocchi et al, 2025). Here we showed how including a broad distribution of production rules results in power-law learning curves for classification, as observed in real data, whereas, surprisingly, it doesn't affect the scaling law of next-token prediction. Extending these results to the case of non-random rules and varying tree topology is an important direction for future studies.

---

> > ### Comment · Reviewer_GeDB · 2025-04-02
> >
> > Thank you for your clarification. Considering both the theoretical elegance and the gap to real-world scaling laws. I have no more questions and would like to retain my rating.

---

### Decision · Program_Chairs · 2025-05-01

**Decision:**

Accept (poster)

**Comment:**

This paper presents a theoretical study of learning curves in neural networks trained on hierarchically structured data with Zipf-distributed features, modeled via a probabilistic context-free grammar. It extends prior work on the Random Hierarchy Model by introducing nonuniform (power-law) production rules. The key finding is that Zipf-distributed rules lead to power-law learning curves in classification, while next-token prediction exhibits a scaling exponent governed solely by the hierarchical structure, independent of Zipf statistics.

Reviewers generally found the theoretical analysis sound and the experimental validation on synthetic data well-executed. The distinction between classification and prediction regimes is novel and insightful. One reviewer raised concerns about clarity and reliance on prior work, suggesting the paper could be made more self-contained. The authors provided a thorough rebuttal, addressing these concerns and outlining plans for improving the manuscript.

Overall, the paper makes a solid theoretical contribution to our understanding of neural scaling laws and the effects of data structure. While some exposition improvements are needed, the results are valuable and relevant to the ICML community.